# Effect of Al Additions and Cooling Rate on the Microstructure and Mechanical Properties of Austenite FeMnAlC Steels

**DOI:** 10.3390/ma15103574

**Published:** 2022-05-17

**Authors:** Cunyu Wang, Chenxing Cao, Jing Zhang, Hui Wang, Wenquan Cao

**Affiliations:** Central Iron & Steel Research Institute (CISRI), Beijing 100081, China; caocx2@163.com (C.C.); 16601306322@163.com (J.Z.); hui051889@163.com (H.W.); caowenquan@nercast.com (W.C.)

**Keywords:** FeMnAlC, austenite, low-density steel, impact absorption energy

## Abstract

The precipitation behavior of κ-carbide and its effects on mechanical properties in Fe-30Mn-xAl-1C (x = 7–11%) steels under water quenching and furnace cooling are studied in the present paper. TEM, XRD, EPMA were employed to characterize the microstructure, and tensile test and the Charpy impact test were used to evaluate mechanical properties. The results show that the density decreases by 0.1 g/cm^3^ for every 1 wt.% of Al addition. The excellent mechanical properties of tensile strength of 880 MPa and impact absorption energy of 120–220 J at −40 °C with V notch were obtained, with both solid solution and precipitation strengthening results in the yield strength increasing by about 57.5 MPa with per 1% Al addition in water-quenched samples. The increasing of yield strength of furnace-cooled samples comes from the relative strengthening of κ-carbides, and the strengthening potential reaches 107–467 MPa. The lower the cooling rate, the easier it is to promote the precipitation of κ-carbides and the formation of ferrite. The partitioning of C, Mn, Al determines the formation of κ-carbides at a given Al addition, and element partition makes the κ-carbides sufficiently easy to precipitate at a low cooling rate. The precipitation of κ-carbides improves strength and does not significantly reduce the elongation, but significantly reduces the impact absorption energy when Al addition ≥ 8%.

## 1. Introduction

Aluminum is often used as a deoxidizer in many low- and medium-alloy steels with a small addition because of its easy oxidation characteristics [1], it is also used as a main alloying element in steels with high alloy content, such as FeMnAlC steel systems [2,3]. Addition of Al decreases the density and Young’s modulus [4,5], increases the strength [6,7], increases the SFE [8], which alters the deformation mechanisms [9,10], increasing the onset for the TRIP and the TWIP effects [11], and changes the hardening mechanism [12,13], affects the corrosion resistance [14,15] and oxidation resistance [16], and so on [17].

Austenite FeMnAlC steels have been paid more and more attention and in-depth study in recent years because of their characteristics of lower density, relatively greater strength, excellent ductility, and toughness [18,19]. Al usually exists in FeMnAlC steel in solid solution and precipitation state—the effect of Al as a solid solution strengthener is 9.3 MPa/wt.% of Al [6], and as precipitate as κ-phase it can be a promising candidate for precipitation hardening of the steels [20]. κ-phase has a perovskite structure of L1_2_, which is coherent with the collective, and the composition range of κ-carbide is (Fe, Mn)_4-y_Al_y_C_x_ (0.8 < y < 1.2, x < 1) [21]. Al affects the precipitation of κ-phase in morphology and precipitation location [22,23], for example, the κ-phase features easy formation also within grain when the Al content exceeds 7% [24], and the addition of Al mainly promotes the precipitation of nanosized κ-carbides and control mechanical properties [25]. The cooling rate also affects the solid solution or precipitation state of Al element in FeMnAlC steel [26]—the lower the cooling rate, the easier the precipitation of κ-phase. We usually want to add more Al to obtain a lower density of steels, and at the same time [27], we inevitably face the problem of cooling rate control after almost all hot working, thus, it is important to know the effect of Al additions and cooling rate on the microstructure and mechanical properties. In the present paper, different amounts of Al were added to 1C-30Mn-xAl-Fe steels and the effect of Al additions and cooling rate on the microstructure and mechanical properties of austenite FeMnAlC steels were studied.

## 2. Experimental Procedures

Fe-30Mn-xAl-1C steels with aluminum content ranging from 7% to 11% (wt. pct.) were melted in a vacuum induction furnace in capacity of 50 kg, ingots were homogenized at 1150 °C and then forged into billet with sectional dimension of 40 mm × 40 mm. Samples for the experiment were prepared from a bar with a diameter of 15 mm forged from the billet.

Density measurement of experiment steels with different aluminum content was carried out according to the Archimedes drainage method. To evaluate the effect of Al on the microstructure and mechanical properties, samples were subjected to solution treatment, which involves heating at 1050 °C for 2 h first, followed by quenching into water at room temperature, or furnace cooling to room temperature. A tensile test was performed on the dog-bone-shaped specimens with gauge length of 25 mm and diameter of 5 mm at a strain rate of 10^−3^/s in an MTS at ambient temperature, and a Charpy impact test was carried out on specimens with a size of 10 × 10 × 55 (mm) and a V-type notch, at a temperature of −40 °C.

Microstructures of specimens were evaluated with transmission electron microscopy (TEM, FEI Tencnai G2 F20), X-ray diffraction (XRD PHILIPS APD-10) and backscatter diffraction (JEOL JSM 7100F). For the microstructure observation in XRD, samples were ground and polished mechanically, and then electro-polished in solution of 7% perchloric acid and 93% alcohol at room temperature, at a current density of 450 mA/cm^2^. For the microstructure observation in TEM, samples were first ground mechanically to a thickness of about 0.04 mm, and then were electro-polished in a twin-jet machine in a solution of 10% perchloric acid and 90% alcohol at about −20 °C, under 25 V voltage.

## 3. Results and Discussion

### 3.1. Density Measurement

The density measurement results of Fe-30Mn-xAl-1C (x = 7–11%) steel with different additions of Al are shown in Figure 1. With the Al content increasing from 7% to 11%, the density of steels decreases linearly from 7.0 g/cm^3^ to 6.6 g/cm^3^. The density decreases by 0.1 g/cm^3^ for every 1 wt.% of Al addition, which is identical to the results in [28].

### 3.2. Microstructure

The microstructure characterized by TEM with different addition of Al, ranging from 7% to 11%, subjected to water quenching or furnace cooling after heating at 1050 °C for 2 h, is shown in Figure 2. The morphology and diffraction pattern results characterized by transmission electron microscopy showed that all samples obtained an austenite matrix, the samples with Al addition of 7% (Figure 2a,b) and 8% (Figure 2c,d) obtained austenite + dislocation microstructures in both water-cooling and furnace-cooling conditions. The sample with Al addition of 9% obtained austenite + dislocation microstructures under water-cooling conditions (Figure 2e), and obtained austenite + κ-carbide while the under the condition of furnace cooling (Figure 2f); the sample with Al addition of 10% subject to water quenching obtained an austenite matrix and disordered κ-carbide structure (Figure 2g), while obtained the coarse-ordered κ-carbide under furnace-cooling condition (Figure 2h); for the sample with Al addition of 11% (Figure 2i,j), a small amount of ferrite appears compared to that with 10% Al addition. The selected area diffraction patterns in red and yellow represent the austenite and κ-carbide, respectively. These results show that the microstructure with austenite as the matrix can be obtained under the composition of Fe-30Mn-xAl-1C (x = 7–11); with the increase of Al content and the decrease in cooling rate, the precipitation of κ-carbide is easier to precipitate.

Figure 3 shows the XRD pattern of the experimental steel after being solutionized at 1150 °C × 2 h and furnace cooling to room temperature. The diffraction peak of κ-phase cannot be clearly detected in the samples with Al content ≤9%, while the diffraction peak of κ (220) is detected in the sample with Al content exceeding 10%, and the diffraction peak of ferrite is also detected in the sample with Al addition of 11%. These results are the same as the TEM characterization, the difference is that the κ-carbide phase is found in a sample with Al addition of 9% through TEM characterization, but not found in XRD characterization; this is because XRD can only detect the phase with a certain volume fraction, κ-carbide is too small a quantity to be detected, and the TEM characterization results strongly proved that the κ-carbide could already be formed during furnace cooling when the Al addition is 9%. By comparing the diffraction peaks of γ (111) of furnace-cooling samples with different Al contents, it can be found that the 2θ value corresponding to the diffraction peak of γ (111) gradually decreases with the increasing of Al addition. According to the Bragg equation, the d value of γ (111) crystal interplanar distance increases continuously, indicating that the content of solid solution Al increases continuously, although the precipitated κ-carbides gradually increase. At the same time, it can also be found that the width of the γ (220) diffraction peak increases with the increase of the Al content, which indicates that the supersaturated austenite phase gradually decomposes into the γ_depl_-phase and the κ-phase during the slow cooling process. It is well known that the lattice constant of the κ-phase is slightly larger than that of austenite, which means that the diffraction peak corresponding to κ appears at a lower Bragg angle, while the lattice constant of the decomposed γ_depl_ is slightly larger than that of the austenite phase, and the corresponding diffraction peak is slightly higher than the parent phase austenite; the lattice constants of the γ_depl_-phase and the κ-phase are close to the parent phase [29], thus, the increase in peak width of γ (220) and γ (111) in the samples with greater Al addition is actually a superposition of the peaks of the decomposition products (γ_depl_- and κ-phase) of the austenite phase. The decomposition behavior of austenite is mutually verified with the TEM characterization results.

### 3.3. Mechanical Properties

The mechanical properties of tensile strength (Rm), yield strength (Rp0.2), total elongation (A), and Charpy impact absorption properties (KV2) of Fe-30Mn-xAl-1C steels with different Al addition subjected to water quenching (WQ) or furnace cooling (FC) after heating at 1050 °C for 2 h are shown in Figure 4.

Figure 4a shows the influence of Al addition and cooling method on the strength of the experimental steel. For samples with water quenching, as the Al addition increases from 7% to 11%, the yield strength of the steel gradually increases from 430 MPa to 700 MPa, while the tensile strength does not change much and remains at about 880 MPa. For samples with furnace-cooling, the yield strength increases significantly from 550 MPa to 950 MPa, with Al addition increasing from 7% to 9%, it does not change significantly with Al addition increasing from 9% to 11%. The change of tensile strength with Al addition is similar to that of yield strength, but the increase is smaller. The tensile and yield strength of water-quenched samples are higher than those of the furnace-cooled ones, the value of the increasing yield strength is the maximum when the amount of Al addition is 9%, about 500 MPa, and the value of the increasing tensile strength is about 50 MPa when the Al addition is 7–8%, and about 200 MPa when the Al addition is 9–11%. The yield strength of water-quenched samples increases from 443 MPa with 7% Al addition to 673 MPa with 11% Al addition, after preliminary calculation, the average addition of 1% Al increases the strength by 57.5 MPa; of course, the strengthening effect comes from both the solid solution and precipitation strengthening of Al elements.

Figure 4b shows the influence of Al addition and cooling method on the impact absorption energy and total elongation of experimental steel. Both furnace-cooled and water-quenched test steels have higher ductility as the Al addition increases from 7% to 11%. When the added amount of Al is less than or equal to 8%, the total elongation of the water-quenched sample and the furnace-cooled sample is not much different, about 50%. When the addition of Al is greater than 8%, the total elongation of the water-quenched samples increases slightly, and the total elongation of the furnace-cooled samples decreases slightly. The impact absorption energy decreases with the increasing of Al addition under both water-quenching and furnace-cooling conditions, compared with water-quenched samples, the impact absorption energy of furnace-cooled sample decreases significantly, especially for the steel with Al addition exceeding 8%. When the Al addition is 8%, the impact absorption energy decreases from 200 J of the water-quenched sample to 45 J of the furnace-cooled sample, and when the Al addition exceeds 9%, the impact absorption energy of the furnace-cooled sample decreases by 95%, less than 10 J.

In summary, under water-quenching conditions, the mechanical properties of high strength combined with good toughness are easy to obtain, even the Al addition reaches 11%, the mechanical property of impact absorption energy of 120 J with tensile strength of 880 MPa can be obtained. Whereas, under furnace cooling, steels are highly strengthened and the toughness decreases sharply, the greater the amount of Al added, the worse the toughness will be. It is easy to note that furnace cooling has little effect on ductility, but significantly reduces the toughness of steel.

## 4. Discussion

### 4.1. Microstructure Evolution Mechanism

The previous test results (Figure 2 and Figure 3) show that when the Al addition of the experimental steel exceeds 9%, the precipitation of κ-carbides becomes more and more obvious in the furnace-cooled samples, and the obvious ferrite phase appears in the steel with Al addition of 11%. The equilibrium phase diagram of 30 Mn sampled at 1000 °C, calculated by Thermal-Calc software, shows that with the same C content, ferrite and κ-phases are easy to form in the equilibrium phase diagram with the increase of Al addition, the ferrite phase may appear when Al addition exceeds 9% in steel with composition of C addition of 1% and Mn addition of 30%. Although there is a difference between the calculated and the experimental value, it can still be inferred that the Al element promotes κ-carbide precipitation and ferrite transformation. In present study, when the Al addition is 9%, disordered fine κ-carbides appeared, and when the Al addition reached 11%, a small fraction of ferrite was detected. During the course of cooling, because of the stability of austenite decreasing, high-temperature austenite decomposition into carbon-depleted austenite γ′ and carbon-rich austenite γ″ without obvious phase interface, as the cooling continues, the C atom partition from γ′ into γ″ follows an uphill diffusion mechanism, γ″ gradually becomes an L12-ordered structure, in which L12 is a nanoscale modulation structure, whose lattice constant is close to that of the austenite phase. With the prolongation of the reaction time, the L12-ordered structure transforms into an ordered rectangular domain stack, and then grows into micron-sized granular κ-carbides; at the same time, carbon-depleted austenite γ′ transforms into α-ferrite.

The amounts of Mn, Al, C element distribution in austenite, κ-carbide, and ferrite phase in the furnace-cooling sample with Al addition of 11% were characterized by EPMA [30], the results are presented in Figure 5. It can be found that the weight percent of C in austenite and κ-carbide exceed the average content of steel of 1.0%, reaching 1.46% and 3.14%, respectively, higher than that in ferrite phase of 0.74%, the amount of Al in austenite is lower than that in κ-carbide and ferrite phase, and the amount of Mn in ferrite is lower than that in austenite and κ-carbide, which proves that the formation of ferrite and κ-carbides follows element partition mechanism. In the present study, furnace cooling provides sufficient time for the partitioning of elements and the formation of ferrite and κ-carbides due to the low cooling rate.

### 4.2. Strengthening and Toughening Mechanisms

According to the mechanical properties results shown in Figure 4, the increasing of Al addition and decreasing of cooling rate promote the precipitation of κ-carbides, which improve the strength and slightly reduce the plasticity, but significantly reduce the impact toughness. It has been reported in many studies that the solid solution of Al improves the strength of FeMnAlC steel [29,31], 1% Al addition in solid solution can increase the strength by 9.3 MPa, κ-carbides significantly improve the strength of steel. Song found that κ-carbides began to precipitate and long-range order aged at 600 °C for 15 min, which has a strong strengthening effect, and like traditional carbide strengthening, such as NbC, TiC, VC, etc., the strengthening potential reaches 100–350 MPa. In the present study, the yield strength of water-quenched samples increased from 443 MPa with 7% Al addition to 673 MPa with 11% Al addition. After preliminary calculation, the average addition of 1% Al increases the strength by 57.5 MPa, compared to water-quenched samples, the yield strength of the furnace-cooled samples significantly increased, the relationships of Al addition and increased strength of steel are 7% Al steel-107 MPa, 8% Al steel-247 MPa, 9% Al steel-467 MPa, 10% Al steel-385 MPa, 11% Al steel-282 MPa, respectively. The increasing of yield strength of furnace-cooled samples stems from the relative strengthening of κ-carbides, the strengthening potential reaches 107–467 MPa, which is higher than that of 100–350 MPa reported by Song.

The experimental steels obtain an austenite matrix; it is well known that austenite has 24 slip systems, which makes it easy for the steel to obtain high ductility. According to the work of Hua et al. [32], for an aluminum content of 6% (SFE of 60 mJ/m^2^, preponderant microband regime), the microstructure at a deformation of 5% starts with planar slip, then as the deformation increases, the dislocations become arrays with similar spacing between them along the main directions. Moreover, the dislocation densities increased without change the slip directions. For the highest aluminum content of 12% (SFE of 96 mJ/m^2^, preponderant microband regime), the microstructure at a deformation of 5% starts with a large number of uniformly arranged slip bands, and then the density of the slip bands increases as the deformation increases until 10%. As deformation proceeded until 30% and fracture, slip bands of high density were found, intersecting each other. In the present study, the work hardening mechanism microband slip was detected in a water-quenched sample with 11% added Al under tensile strain of 10% (Figure 6a), the phenomenon of dislocation cutting through κ-carbides was also detected in the furnace-cooled sample (Figure 6b), which is similar to the work of Morris et al. [33] and Gutierrez et al. [34]. Thus, the work hardening mechanism dominated by dislocation slip determines the high ductility of the experimental steels.

The impact energy of furnace-cooled samples dramatically reduces compared to that of water-quenched samples, the fracture morphology characterized by SEM is given in Figure 7. The water-quenched sample has a dimple-dominated fracture morphology, while the furnace-cooled sample has a cleavage fracture morphology. According to the results of microstructure characterization, it can be found that the precipitation of fine and disordered κ-phase has a relatively small effect on toughness, while the ordered and coarse k-phase significantly reduces the impact absorption energy. The results of the study on the precipitation mechanism of κ-phase show that the formation mechanism of κ-carbide is the ordering transformation of the matrix caused by amplitude modulation decomposition, and the κ-carbide has a coherent crystallographic relationship with the matrix, [100]κ//[100]γ and [010]κ//[010]γ, κ-carbides appear as flat cuboid structures in three dimensions. When the current size exceeds a certain critical value, it will be very beneficial to the expansion of cracks and form cleavage facets, thereby significantly reducing the impact toughness of the steel.

Thus, compared to the water-quenched experimental steel, the work hardening mechanism dominated by dislocation slip makes steels obtain high ductility and the nucleation and growth of κ-carbide significantly affect toughness in furnace-cooling steels.

## 5. Conclusions

The effects of Al addition and cooling rate on the microstructure and mechanical properties were investigated based on the 1C30Mn(7–11)AlFe steels via water quenching and furnace cooling; the results are summarized as follows:(1)The higher the Al addition, the lower the cooling rate, the easier it is to promote the precipitation of κ-carbides and the formation of ferrite;(2)The partitioning of an element determines the formation of κ-carbides at a given Al addition and element partition sufficiently, which makes the κ-carbides easy to precipitate at a low cooling rate;(3)The precipitation of κ-carbides improves strength and does not significantly reduce the elongation, but it significantly reduces the impact absorption energy when Al addition ≥ 8%;(4)The increasing of yield strength of furnace-cooled samples comes from the relative strengthening of κ-carbides, and the strengthening potential reaches 107–467 MPa.

## Figures and Tables

**Figure 1 materials-15-03574-f001:**
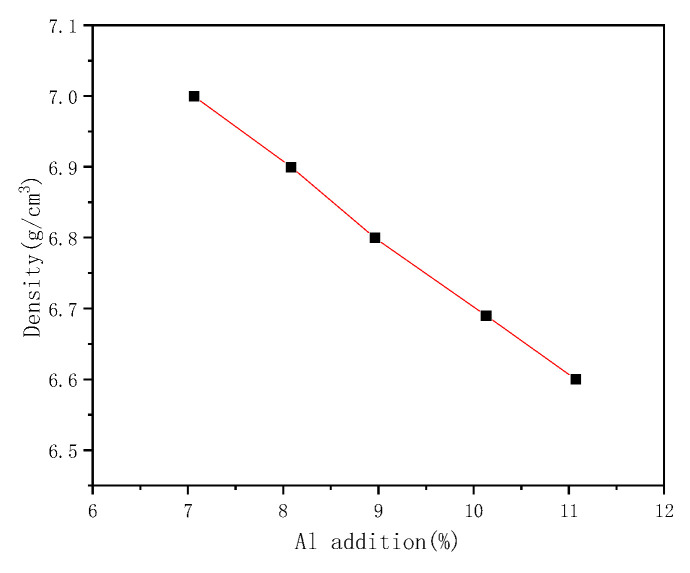
Effect of Al addition on the density reduction of Fe-30Mn-xAl-1C steel.

**Figure 2 materials-15-03574-f002:**
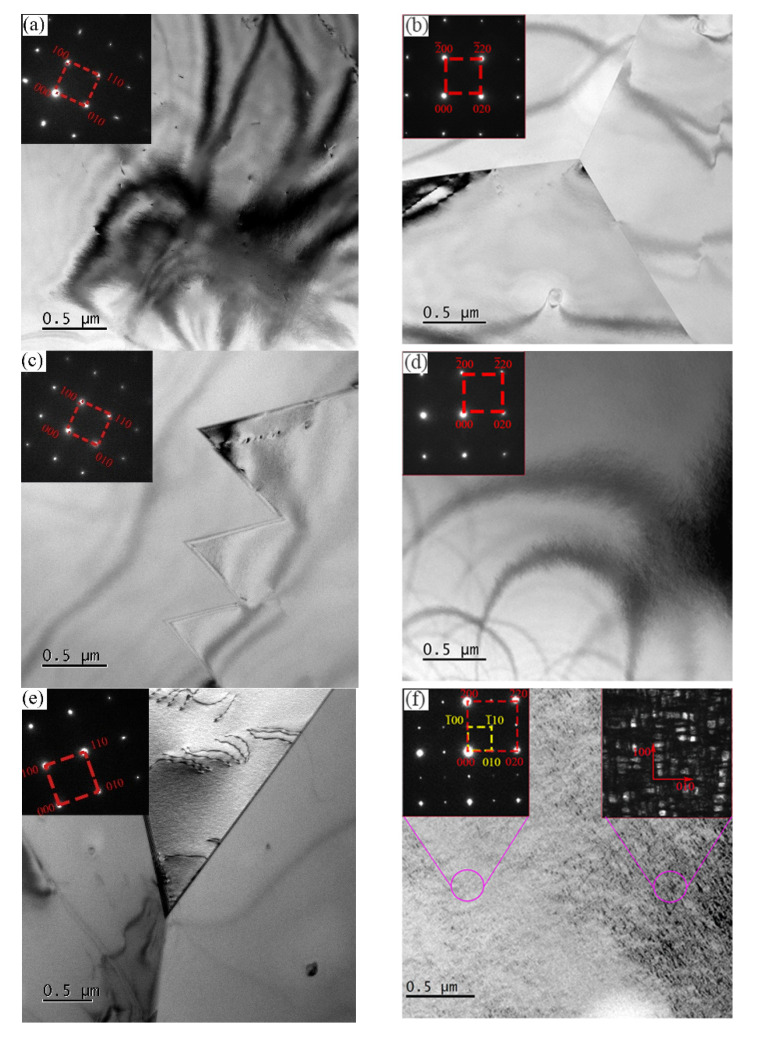
TEM morphology and diffraction patterns of experimental steels: (**a**) 7% Al under water-quenched, (**b**) 7% Al under furnace-cooled, (**c**) 8% Al under water-quenched, (**d**) 8% Al under furnace-cooled, (**e**) 9% Al under water-quenched, (**f**) 9% Al under furnace-cooled, (**g**) 10% Al under water-quenched, (**h**) 10% Al under furnace-cooled, (**i**) 11% Al under water-quenched, (**j**) 11% Al under furnace-cooled conditions.

**Figure 3 materials-15-03574-f003:**
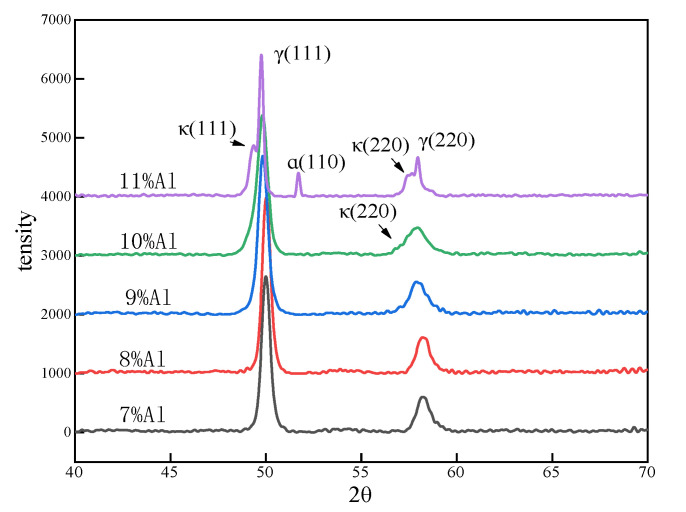
X-ray diffraction of furnace-cooling samples with different Al addition.

**Figure 4 materials-15-03574-f004:**
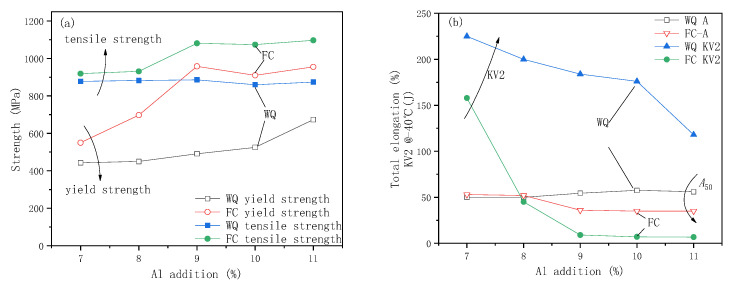
Effect of Al addition on the strength (**a**) and total elongation and Charpy impact absorption properties (**b**).

**Figure 5 materials-15-03574-f005:**
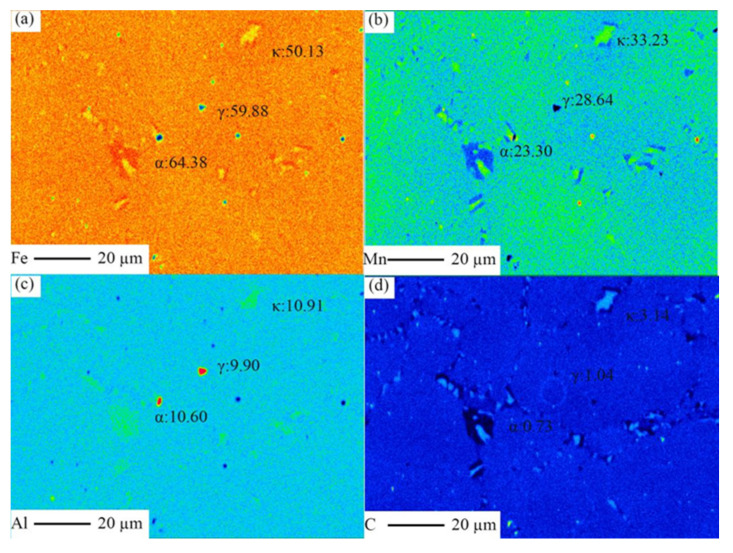
Fe (**a**), Mn (**b**), Al (**c**), and C (**d**) element distribution in a furnace-cooling sample with Al addition of 11% were characterized by EPMA.

**Figure 6 materials-15-03574-f006:**
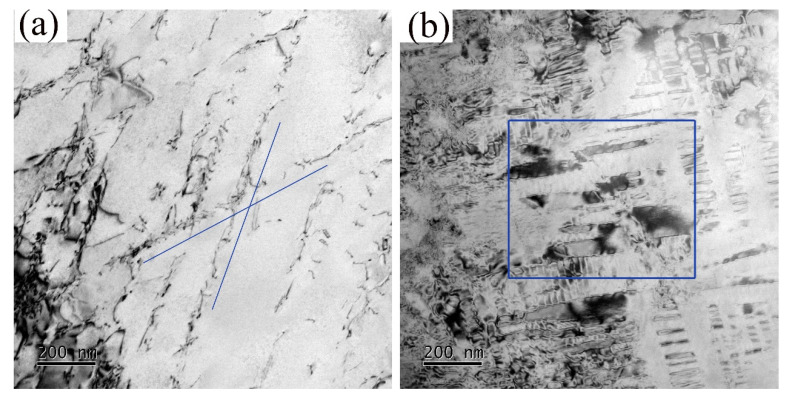
The TEM structure of water-quenched (**a**) and furnace-cooled (**b**) samples with 11% Al addition after tensile strain of 10%.

**Figure 7 materials-15-03574-f007:**
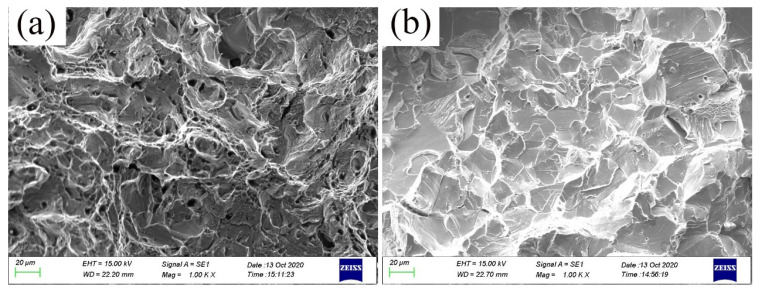
The Charpy impact fracture morphology characterized by SEM of samples with 11% Al addition under water quenching (**a**) and furnace cooling (**b**).

## Data Availability

Not applicable.

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
