# Peer review of "Effect of Al Additions and Cooling Rate on the Microstructure and Mechanical Properties of Austenite FeMnAlC Steels"

_materials, 2022, doi:10.3390/ma15103574_

Round 1

Reviewer 1 Report

A study of the microstructure and mechanical properties of Fe-Mn-Al-C alloys with varying %Al has been presented.  The paper is adequate for publication, although the results do not have a high degree of novelty.  Specific comments are given below:

1) It would be helpful if the authors provided (in the introduction) a description of the structure, composition and space group for the kappa-carbide.

2) Line 73 has subsection 3.3 – there does not appear to be a section 3.2.

3) Line 77 states “all samples obtained a martensitic matrix” – is this correct? Please clarify.

4) Line 199 – please specify compositions as weight or atomic percent, as appropriate.

5) Line 212, where it is written, “It has been reported in many references” – please give specific references

6) Line 264 – the word “nuclear” should be “nucleation”

7) Figure 5- the writing inside the figures is too small and difficult to read

8) Reference 2 – give date/issue/pages, etc.

9) The word “Sharpy” in the paper should be spelled “Charpy”

Reviewer 2 Report

The idea of the work "Effect of Al additions and cooling rate on the microstructure and mechanical properties of austenite FeMnAlC steels" and its structure, according to the reviewer, deserve a positive assessment and can be published in the journal after a minor revision. 

Please pay attention to the following points:

- I think it is necessary to substantiate the relevance of the study in more detail in the introduction.

- Were all samples obtained using the same technology? I ask you to describe the deformation-thermal processing modes. I understand that all samples were quenched before testing, but the background of the material is important to the reader.

- The red color in figure 2 is shown poorly.

- Different scale bars are used in different compositions (figure 2), which are compared with each other.

- SAED patterns require more attention (figure 2)

- The article does not explain what may be the reason for the change in the width of the X-ray lines 111 220 in Figure 3

 -In the conclusions, it is necessary to indicate the range of aluminum content that was investigated

There are many mistakes in the format of references

- The discussion is well and persuasive written. However, contains few references to already known references, few comparison elements
